# A conserved interaction of the dynein light intermediate chain with dynein-dynactin effectors necessary for processivity

In-Gyun Lee[1], Mara A. Olenick[1], Malgorzata Boczkowska[1], Clara Franzini-Armstrong[2], Erika L.F. Holzbaur[1] & Roberto Dominguez[1]

Cytoplasmic dynein is the major minus-end-directed microtubule-based motor in cells. Dynein processivity and cargo selectivity depend on cargo-specific effectors that, while generally unrelated, share the ability to interact with dynein and dynactin to form processive dynein–dynactin-effector complexes. How this is achieved is poorly understood. Here, we identify a conserved region of the dynein Light Intermediate Chain 1 (LIC1) that mediates interactions with unrelated dynein–dynactin effectors. Quantitative binding studies map these interactions to a conserved helix within LIC1 and to N-terminal fragments of Hook1, Hook3, BICD2, and Spindly. A structure of the LIC1 helix bound to the N-terminal Hook domain reveals a conformational change that creates a hydrophobic cleft for binding of the LIC1 helix. The LIC1 helix competitively inhibits processive dynein–dynactin-effector motility in vitro, whereas structure-inspired mutations in this helix impair lysosomal positioning in cells. The results reveal a conserved mechanism of effector interaction with dynein–dynactin necessary for processive motility.

[1] Department of Physiology, Perelman School of Medicine, University of Pennsylvania, Philadelphia, PA 19104, USA. [2] Department of Cell and Developmental Biology, University of Pennsylvania, Philadelphia, PA 19104, USA. Correspondence and requests for materials should be addressed to R.D. (email: droberto@pennmedicine.upenn.edu)

Cytoplasmic dynein 1 (dynein) is the major minus-end-directed microtubule-based motor in eukaryotic cells. It is responsible for the transport of very diverse cargoes from the periphery to the center of the cell, including lysosomes, mitochondria and autophagosomes[1–4]. Recent work has shown that both cargo-specificity and processivity depend on the interaction of dynein with its general adaptor, the dynactin complex, and a series of cargo-specific effectors, including BICD2[5–7], Hook1/3[5,8,9], Spindly[5], FIP3[5] and NIN/NINL[10]. These proteins are generally unrelated at the sequence level, but they all contain large portions of predicted coiled-coil structure and share the ability to interact with both dynein and dynactin to activate processive motility[5,6,8,10,11]. It remains unclear, however, whether each effector has evolved these functions independently or whether they share common structural-functional features and similar interactions with dynein and dynactin. Here, we show that a conserved amphipathic helix within the unstructured C-terminal region of the dynein Light Intermediate Chain 1 (LIC1) interacts with diverse dynein–dynactin effectors. The interactions were quantitatively characterized using purified proteins and isothermal titration calorimetry (ITC). A crystal structure of the LIC1 helix in complex with the N-terminal Hook domain of Hook3 reveals a conformational change within the Hook domain that gives rise to a hydrophobic cleft where the LIC1 helix binds. Supporting the importance of the LIC1-effector interaction, we found that the LIC1 helix competitively inhibits the processive motility of dynein–dynactin in complex with either Hook3 or BICD2 in single-molecule assays using total internal reflection fluorescence (TIRF) microscopy. Finally, in cellular assays, mutating the LIC1 helix leads to defective dynein-driven positioning of lysosomes. Together, the results reveal the existence of a conserved mechanism of interaction between functionally unrelated dynein–dynactin effectors and the dynein LIC1, which is required for processive dynein-driven transport.

## Results

**Hook interacts with the dynein LIC1 via the Hook domain.** The dynein LICs, comprising two closely related isoforms (LIC1 and LIC2), consist of two domains—an N-terminal GTPase-like domain that interacts with the dynein heavy chain[11] and a less conserved and predicted unstructured C-terminal region, referred to here as the effector-binding domain. Using pull-down studies, it had been previously shown that the LIC1-effector-binding domain interacts with several dynein–dynactin effectors, including Hook3, FIP3, BICD2, and Spindly[9,11,12]. On the other hand, a group of dynein-binding proteins, including BICD2, Spindly, HAP1, and TRAK share a coiled-coil segment, termed the CC1-Box, that has been directly implicated in LIC1 binding[12]. Here, we set out to specifically map and quantitatively characterize the interactions of LIC1 with several dynein–dynactin effectors, including Hook1, Hook3, BICD2, and Spindly.

Hook1 and Hook3 are known dynein effectors[5,8,9] that function in endosomal transport[13–15]. We expressed truncated constructs of human Hook1 and Hook3 in *E. coli*, whereas full-length Hook1 was expressed in insect cells (Fig. 1a, b). Because Hook contains several regions of predicted coiled-coil (CC1–4) (Fig. 1a), we first analyzed whether these constructs were dimeric or monomeric using light scattering. A construct corresponding to the N-terminal Hook domain (Hook1$_{11-166}$) was monomeric, whereas construct Hook1$_{11-443}$, extending to the end of CC2 was dimeric (Fig. 1c). In contrast, Hook1$_{11-238}$, comprising only the globular Hook domain and CC1 region, was in equilibrium between dimers and monomers, as indicated by an experimentally measured mass of 39.7 kDa, i.e., intermediate between those of the dimer and the monomer. Full-length human LIC1 was expressed as a fusion protein with MBP to increase its solubility, and was also found to be monomeric by light scattering (Fig. 1c).

Using ITC, MBP-LIC1$_{FL}$ bound Hook1$_{11-443}$ with low micro-molar affinity ($K_D$ = 8.1 μM) and ~1:1 stoichiometry, i.e., two LIC1 molecules per Hook1 dimer (Fig. 1d). Note that this ITC titration was performed at 30 °C, instead of 20 °C for most titrations performed here, because the amount of heat given off by this reaction was too small to allow for reliable fitting of the thermodynamic parameters. Consistent with the light scattering results, the titration of Hook1$_{11-238}$ into buffer produced a significant endothermic reaction, which was interpreted as indicative of dimer dissociation, with a $K_D$ of 2.1 μM (Fig. 1e). This conclusion was confirmed by analysis of Hook1$_{1-239}$GCN4, a dimeric construct stabilized through the addition of a GCN4 leucine zipper at the C terminus, whose titration into buffer did not produce any significant heat change (Fig. 1e). Hook1$_{1-239}$GCN4-bound MBP-LIC1$_{FL}$ with a $K_D$ of 12.9 μM and ~1:1 stoichiometry (Fig. 1f), which is very similar to what was observed with Hook1$_{11-443}$ (Fig. 1d) despite the fact that the titration was inverted by placing Hook1$_{1-239}$GCN4 in the syringe and MBP-LIC1$_{FL}$ in the cell. The monomeric construct Hook1$_{11-166}$ also bound MBP-LIC1$_{FL}$ with similar affinity ($K_D$ = 12.7 μM) and 1:1 stoichiometry. Together these results show that: (a) the LIC1-binding site is fully contained within the conserved N-terminal Hook domain, (b) each Hook dimer interacts with two LIC1s, and (c) the CC1 region of Hook forms an unstable coiled-coil, which on its own cannot support stable Hook dimerization.

To gain further insights into the overall structure of Hook and the disposition of the Hook domain with respect to the coiled-coil segments, we used rotary shadowing electron microscopy to visualize full-length Hook1 (Fig. 1h). Hook had a kinesin-like appearance, with most particles displaying two well-separated globular domains at one end, connected through a short neck-like region to a long thin rod, which was often interrupted by a pronounced kink, followed by a shorter thin rod. These features were interpreted to correspond to the N-terminal Hook domain, the unstable CC1 region, CC2, the central so-called Spindly motif[12], and CC3, respectively (Fig. 1i). The smaller C-terminal cargo-binding domain (CBD) was only occasionally visualized as a defined structural feature (Fig. 1i). This assignment of domains is consistent with the length of the segment extending from the end of the neck region to the central kink, whose mean length of

**Fig. 1** Hook interacts with LIC1 via the N-terminal Hook domain. **a** Domain organization of Hook1 and constructs used in this study (CC: coiled-coil, CBD: cargo-binding domain). **b** SDS-PAGE (4–12%) showing several of the proteins used in this study. **c** SEC-MALS analysis of Hook1 constructs and MBP-LIC1$_{FL}$ (color coded as indicated). The molar mass determined from light scattering (right *y*-axis) and the UV absorption at 280 nm (left *y*-axis) are plotted as a function of the elution volume. The theoretical masses are given in parenthesis. **d–g** ITC titrations of LIC1 and Hook1 constructs as indicated. Listed with each titration are the concentrations of the protein in the syringe and in the cell, as well as the temperature of the experiment and parameters of the fit (stoichiometry *N*, dissociation constant $K_D$). Errors correspond to the s.d. of the fits. Open symbols correspond to titrations into buffer (except part **e**, where both titrations are into buffer). **h** Representative rotary shadowing EM image of Hook1$_{FL}$. Scale bar, 100 nm. White squares indicate individual Hook1$_{FL}$ molecules highlighted in the zooms shown on the right. Scale bar, 50 nm. **i** Close-up view of a representative Hook1$_{FL}$ molecule shown alongside a cartoon representation of the Hook1 domains based on the rotary shadowing EM, secondary structure and sequence conservation analyses (see Supplementary Fig. 1). Scale bar, 50 nm. **j** Length distribution of the region spanning from the end of the neck to the kink. Bin size, 5 nm, *n* = 33

~31 nm approximately corresponds to the predicted dimensions of CC2 (~27 nm) (Fig. 1j), as estimated from the structures of other coiled-coil proteins. The dimensions of the remaining smaller domains cannot be accurately measured at this resolution. The assignment of domains is also consistent with structural predictions and sequence conservation analyses, showing a series

of coiled-coil segments (CC1, CC2, CC3, and CC4) interspersed with three globular regions (Hook domain, Spindly motif, and CBD), connected by short, unstructured loops of lower sequence conservation (Supplementary Fig. 1). The variability of the kink angle between CC2 and CC3 suggests that the regions N- and C-terminal to the Spindly motif move relatively independently of

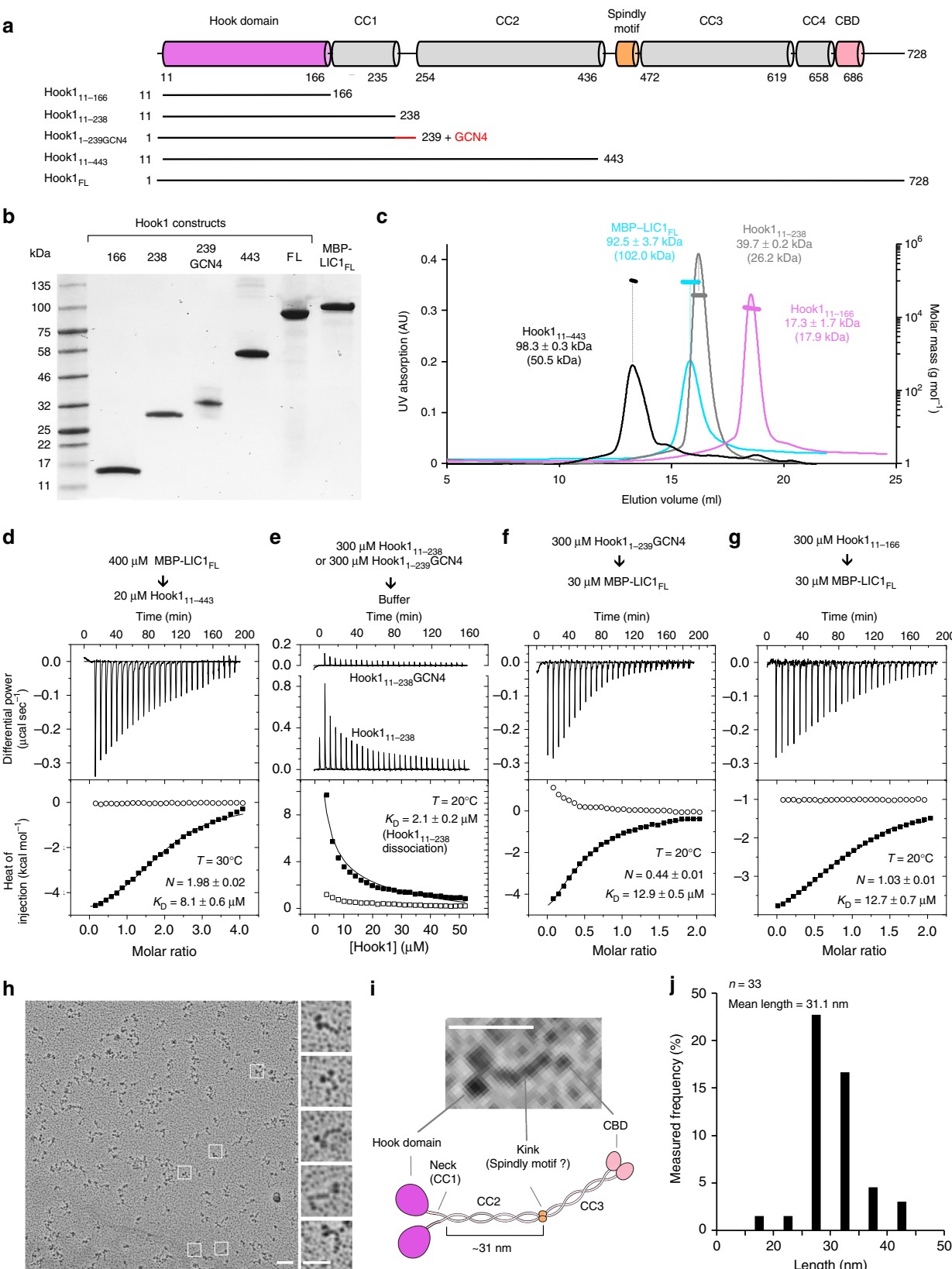

each other, i.e., the Spindly motif appears to function as a 'hinge'. Finally, the fact that the two globular Hook domains appear well separated from each other in most of the particles visualized is consistent with the two helices that form the CC1 segment (neck) not forming a stable coiled-coil, as also suggested by the light scattering (Fig. 1c) and ITC (Fig. 1e) results.

**A helix in LIC1 C-terminal region binds the Hook domain.** The C-terminal effector-binding domain of LIC1 (human LIC1 residues 390–523) has been shown to interact with dynein–dynactin effectors, including Hook3, BICD2, and Spindly[9,11,12]. However, it was unknown whether different effectors bound to the same or different regions on the LIC1 C terminus, and these interactions

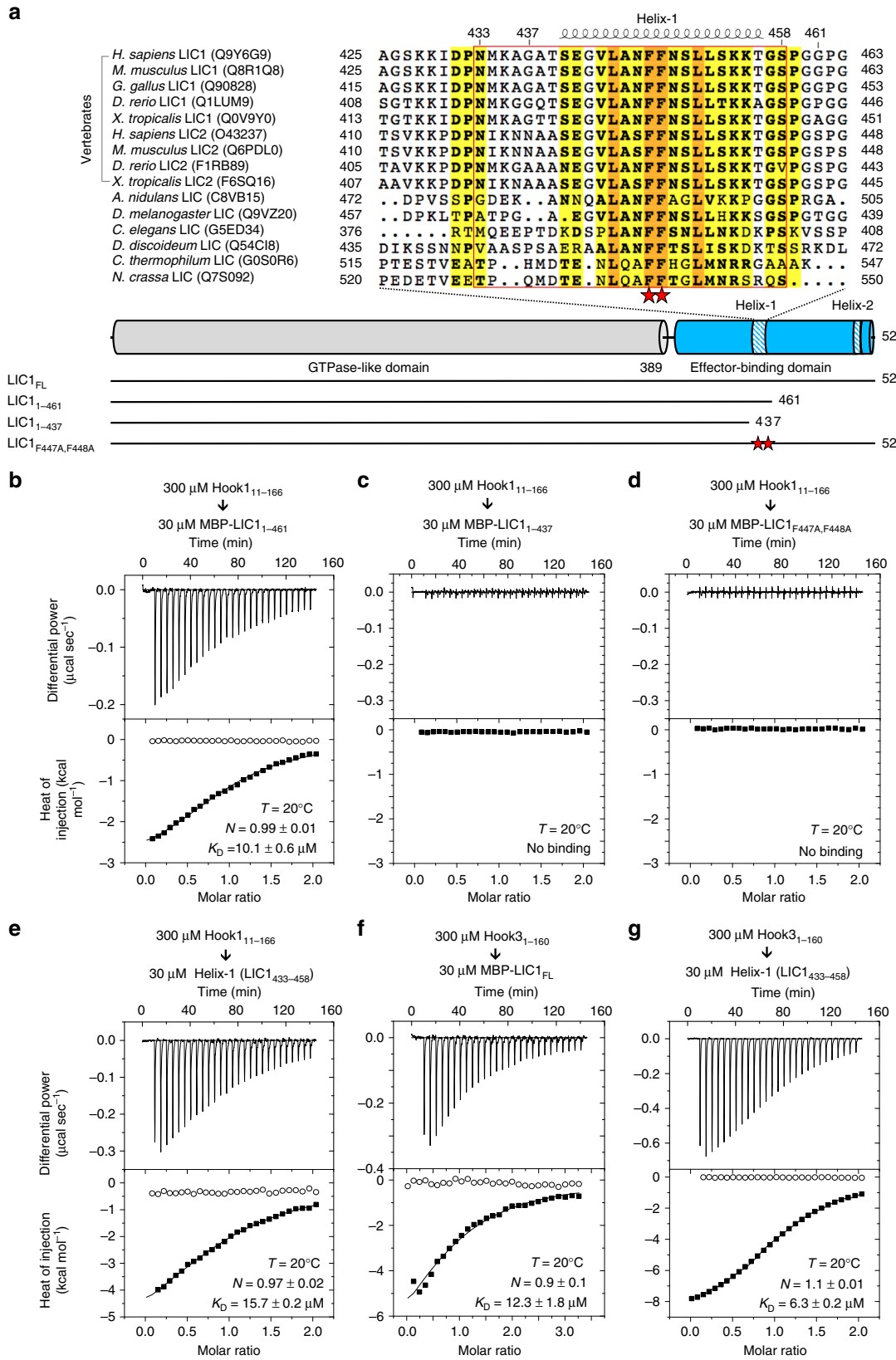

were characterized by qualitative rather than quantitative analyses. Here, we set out to map the specific region of the LIC1 C terminus implicated in interactions with Hook and other effectors (see below) and quantitatively characterize the interactions. Most of the LIC1-effector-binding domain is predicted to be unstructured and, unlike the GTPase-like domain, it is not highly conserved among species (Supplementary Fig. 2). However, sequence analysis reveals two regions of relatively high conservation that coincide with predicted α-helical segments, which we named Helix-1 (human LIC1 residues 440–456) and Helix-2 (residues 493–502) (Fig. 2a and Supplementary Fig. 2). To test whether these conserved helical segments participate in the interaction with the Hook domain, we generated two C-terminally truncated LIC1 constructs, MBP-LIC1$_{1-461}$, which removes the region C-terminal to Helix-1 and MBP-LIC1$_{1-437}$, which additionally removes Helix-1 (Fig. 2a). MBP-LIC1$_{1-437}$ failed to bind the Hook domain by ITC (Fig. 2c), whereas MBP-LIC1$_{1-461}$ bound the Hook domain (Fig. 2b) with nearly the same affinity ($K_D = 10.1$ μM) as MBP-LIC1$_{FL}$ ($K_D = 12.7$ μM) (Fig. 1g). These results suggested that the binding site is contained within Helix-1. Consistent with this conclusion, the Hook domain failed to bind to construct MBP-LIC1$_{F447A,F448A}$, in which two strictly conserved phenylalanine residues in the middle of Helix-1 were simultaneously mutated to alanine (Fig. 2d).

To further test the role of Helix-1 in Hook binding, we expressed a 26-a.a. peptide (LIC1$_{433-458}$), extending several amino acids N- and C-terminally to the predicted helical segment to ensure proper folding of Helix-1. The Hook domain of Hook1 bound to LIC1$_{433-458}$ with nearly the same affinity ($K_D = 15.7$ μM) as to MBP-LIC1$_{FL}$ (Figs. 1g and 2e). The Hook domain of Hook3 (human Hook3 residues 1–160) also bound MBP-LIC1$_{FL}$ and Helix-1 with similar affinities (Fig. 2f,g), and these affinities were comparable to those observed with the Hook domain of Hook1 (Figs. 1g and 2e). Together, these results map the LIC1–Hook interaction to the conserved Helix-1 within the effector-binding domain of LIC1 and the N-terminal Hook domain of both Hook1 and Hook3. Furthermore, the conserved hydrophobic residues F447 and F448 within Helix-1 likely form part of the binding interface.

**Structure of a complex of the Hook domain and the LIC1 helix**. To further understand the mechanism of interaction between Hook and LIC1, we determined the crystal structure of human Hook3$_{1-160}$ in complex with human LIC1 Helix-1 at 1.5 Å resolution (Fig. 3a–c and Table 1). The electron density is well defined for Hook3 residues 10–160 and LIC1 residues 441–454 (Fig. 3b). The first nine amino acids of Hook3 and residues 433–440 and 455–458 of Helix-1 were disordered and are, thus, unlikely to participate in the interaction. As previously reported[9], the Hook domain displays a canonical 7-helix calponin homology (CH)-like fold, featuring an additional helix at the C terminus termed helix α8. Generally, the structure superimposes well with that of the unbound Hook domain determined previously[9], with an r.m.s. deviation of 1.4 Å for 136 equivalent Cα atoms (Fig. 3d). However, the Hook domain-specific helix α8, which in the

unbound structure is fully extended and interacts in anti-parallel fashion with the same helix from a symmetry-related molecule in the crystal, is broken into two helices (α8a and α8b) in the current structure (Fig. 3d), giving rise to a V-shaped hydrophobic cleft that constitutes the binding site for LIC1 Helix-1 (Fig. 3c). As predicted, the visualized portion of Helix-1 is folded as an amphipathic α-helix, with its hydrophobic surface facing the hydrophobic cleft of the Hook domain (Fig. 3c). All the highly conserved, hydrophobic amino acids of the LIC1-effector-binding domain are directly inserted into the hydrophobic cleft of the Hook domain, including L444, F447, F448, and L451, explaining why the mutant MBP-LIC1$_{F447A,F448A}$ failed to bind the Hook domain (Fig. 2d). The binding interface also coincides with the most highly conserved surface of the Hook domain (Fig. 3e).

The Hook domain also interacts with a second LIC1 Helix-1 from a neighboring complex in the crystal lattice (Fig. 3e). This interaction presents the less conserved, hydrophilic surface of Helix-1 to a less conserved surface on the Hook domain, which a priori is inconsistent with a native interaction. Yet, to rule out this interaction, we generated two Hook domain mutants: A138D, testing the presumed crystal packing contact, and M140D, testing the anticipated native binding site (Fig. 3f). The Hook3$_{1-160}$A138D mutant bound MBP-LIC1$_{FL}$ with the same affinity as wild type Hook3$_{1-160}$ (compare Figs. 2f and 3g), whereas the Hook3$_{1-160}$M140D mutant failed to bind MBP-LIC1$_{FL}$ (Fig. 3h), confirming that the native binding site of Helix-1 is located at the interface between α8a and α8b, and conferring functional significance to the conformational change that splits helix α8 into two helices. Indeed, even in the presence of Helix-1, we obtained a second crystal form showing the reported extended conformation of helix α8[9], but the LIC1 peptide was not bound in these crystals. To further test the importance of the conformational change in helix α8 for LIC1 binding, we generated a truncated construct, Hook3$_{1-143}$, lacking the α8b portion of helix α8, i.e., the region that bends back to form the V-shaped cleft (Fig. 3d). Hook3$_{1-143}$ failed to bind MBP-LIC1$_{FL}$ (Fig. 3i). Collectively, these results confirm that the extended helix α8 of the Hook domain, which distinguishes this domain from the canonical CH fold, undergoes a conformational change to produce a conserved, hydrophobic cleft for binding of the conserved LIC1 Helix-1.

**The LIC1 helix binds diverse dynein–dynactin effectors**. Next, we asked whether LIC1 Helix-1 was also implicated in interactions with other dynein–dynactin effectors that are generally structurally and functionally unrelated to each other. As mentioned above, a recent study found that a group of dynein–dynactin effectors share a region termed the CC1-Box that was implicated in LIC1 binding through pull-down and mutagenesis studies[12].

To test whether LIC1 Helix-1 also mediates the interaction with CC1-Box-containing effectors, we expressed N-terminal fragments of two effectors: BICD2$_{1-98}$ and Spindly$_{1-142}$ (Fig. 4a, b). These constructs extend N- and C-terminally from the CC1-Box to include the first predicted coiled-coil segment of each

**Fig. 2** The conserved Helix-1 within the LIC1-effector-binding domain binds the Hook domain. **a** Alignment of LIC sequences from different species and isoforms around the predicted Helix-1 within the C-terminal effector-binding domain (top) and domain diagram of human LIC1 showing the constructs used in this study (bottom). The name of each sequence includes the organism of origin and UniProt accession code. Yellow and orange backgrounds indicate 70% and 100% sequence conservation, respectively. Red stars highlight residues F447 and F448 that were mutated to alanine. The predicted Helix-1 and Helix-2, coinciding with regions of higher sequence conservation (see Supplementary Fig. 2), are highlighted in the domain diagram, and Helix-1 is also depicted above the sequence alignment. The region corresponding to the Helix-1 (LIC1$_{433-458}$) peptide is contoured red. **b–g** ITC titrations of Hook1$_{11-166}$ and Hook3$_{1-160}$ into LIC1 constructs (as indicated). Listed with each titration are the concentrations of the protein in the syringe and in the cell, as well as the temperature of the experiment and parameters of the fit (stoichiometry N, dissociation constant $K_D$). Errors correspond to the s.d. of the fits. Open symbols correspond to control titrations into buffer

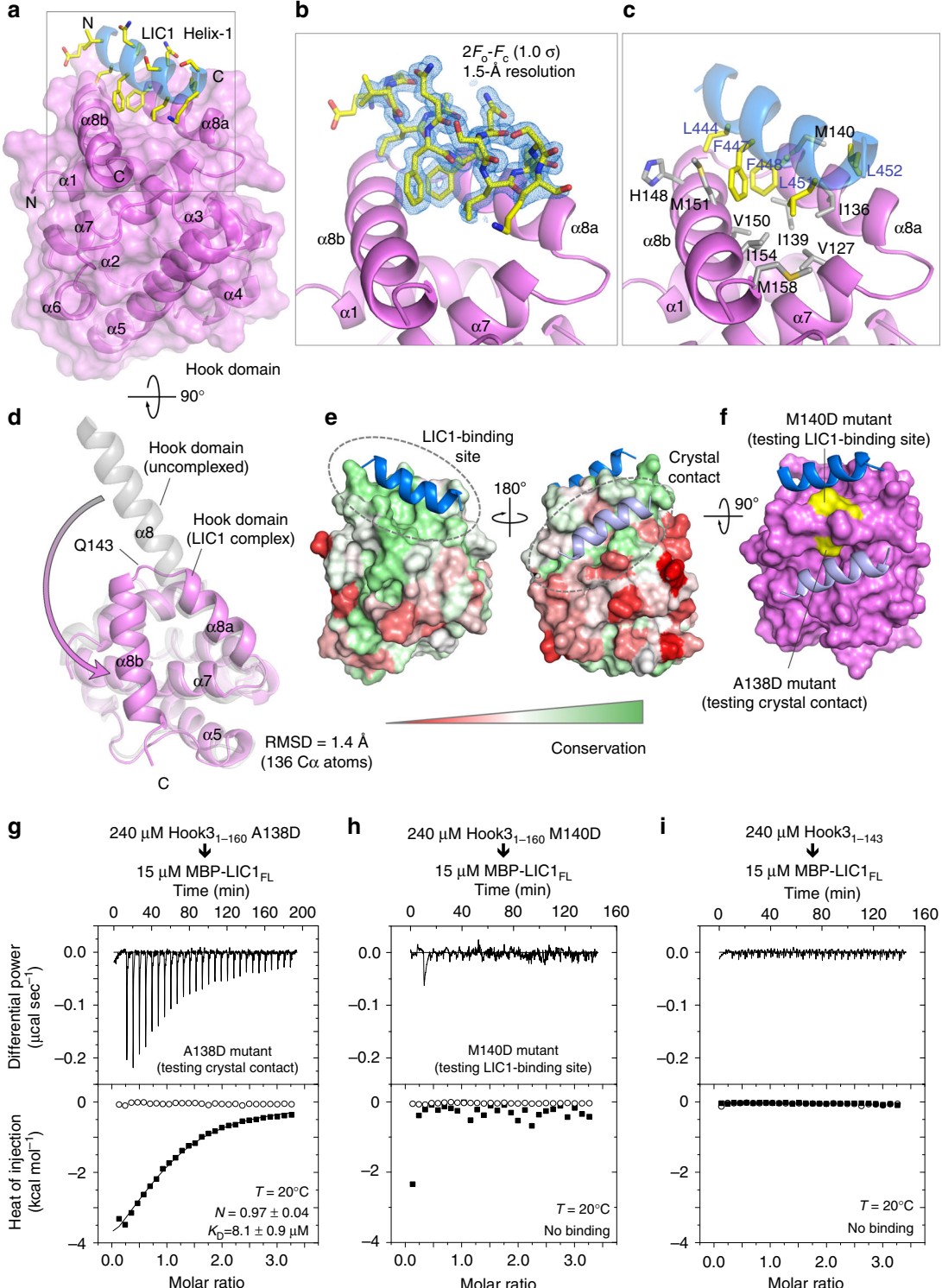

**Fig. 3** Crystal structure of the Hook domain in complex with LIC1 Helix-1. **a** Ribbon and surface representation of the structure of Hook3$_{1-160}$ (magenta) in complex with Helix-1 (LIC1$_{433-458}$, blue). The side chains of Helix-1 are shown using a sticks representation, colored by atom type. **b** Close-up view of the Helix-1 binding site, showing the 2$Fo$-$Fc$ electron density map (blue mesh) at 1.5 Å resolution, contoured at 1 σ around an all-atom representation of Helix-1. **c** Close-up view of the Helix-1 binding site, showing the residues at the hydrophobic contact interface. **d** Superimposition of the structure of the Hook domain from the Helix-1-bound complex (magenta) and unbound structure (gray)[9]. A conformational change in the C-terminal helix α8, which distinguishes this domain from the CH domain, leads to the formation of two helices (α8a and α8b) that constitute the binding site for Helix-1. **e** Sequence conservation of the Hook domain (see also Supplementary Fig. 1) mapped onto the surface of the structure and colored from low to high conservation using a red to green gradient. In the crystal lattice, the Hook domain contacts a second Helix-1 from a neighboring complex (light blue). **f** Surface representation of the Hook domain (magenta), showing in yellow the two amino acids mutated (A138D and M140D) to test the functional relevance of the two Helix-1 interactions. **g**–**i** ITC titrations of the indicated Hook3$_{1-160}$ mutants into MBP-LIC1$_{FL}$. Experimental conditions and fitting parameters are listed. Errors correspond to the s.d. of the fits. Open symbols correspond to titrations into buffer

**Table 1 Data collection and refinement statistics**

| Data collection | |
|---|---|
| Space group | $P\,2_1\,2_1\,2_1$ |
| Cell dimensions | |
| $a, b, c$ (Å) | 31.71, 35.37, 126.14 |
| $\alpha, \beta, \gamma$ (°) | 90.0, 90.0, 90.0 |
| Resolution (Å) | 50.00–1.50 (1.55–1.50)[a] |
| $R_{merge}$ | 0.053 (0.373) |
| $I / \sigma I$ | 50.7 (3.8) |
| Completeness (%) | 98.7 (90.9) |
| Redundancy | 11.2 (5.6) |
| Wilson $B$-factor (Å$^2$) | 14.1 |
| **Refinement** | |
| Resolution range (Å) | 20.54–1.50 (1.55–1.50) |
| No. of reflections | 23,302 |
| $R_{work} / R_{free}$ (%) | 15.5 (18.8) / 17.7 (23.7) |
| No. of non-hydrogen atoms | |
| Protein | 1346 |
| Water | 121 |
| **B-factors (Å$^2$)** | |
| Protein | 19.4 |
| Water | 30.9 |
| **r.m.s. deviations** | |
| Bond lengths (Å) | 0.015 |
| Bond angles (°) | 1.490 |
| **Ramachandran (%)** | |
| Favored | 97 |
| Outliers | 0 |
| PDB code | 6B9H |

The dataset was collected from a single crystal at CHESS beamline F1
[a]Values in parentheses correspond to the highest resolution shell

protein and ensure proper dimerization (note that when bound to dynein–dynactin, the entire N-terminal region of these two proteins appear to form uninterrupted coiled-coil structures[16], as depicted in Fig. 4a.) Indeed, as verified by light scattering (Fig. 4c), both BICD2$_{1-98}$ and Spindly$_{1-142}$ form stable coiled-coil dimers, with experimentally determined masses approximately double those calculated from sequence. By ITC, the titrations of LIC1$_{433-458}$ into BICD2$_{1-98}$ (Fig. 4d) and Spindly$_{1-142}$ (Fig. 4e) fitted best to two binding-site isotherms. The affinities of the two binding sites were similar to each other, and they were also similar for the two effectors (with $K_D$s ranging from 1.5 to 7.6 µM). Curiously, however, despite sharing a similar CC1-Box and displaying similar affinities for LIC1 Helix-1, the titrations into BICD2$_{1-98}$ and Spindly$_{1-142}$ had different overall appearances (Fig. 4d, e). For the Spindly$_{1-142}$ titration, in particular, the two binding sites have very close affinities and are probably saturated at the same time, but the first site has a mild endothermic character, whereas the second site has a strong exothermic character, which masks the endothermic signal of the first part of the titration, explaining the peculiar shape of this reaction. Likely, LIC1 binding produces different types of conformational changes in these two proteins, which other than the CC1-Box share no apparent sequence similarity. These results confirm that LIC1 Helix-1 constitutes a common binding site for unrelated dynein–dynactin effectors, including CC1-Box-containing effectors (BICD, Spindly) and Hook-family effectors.

**The LIC1 helix/effector interaction is crucial for motility**. To test the functional significance of the LIC1 Helix-1 interaction with dynein–dynactin effectors, we utilized an in vitro single-molecule approach to track the movement of dynein–dynactin-effector complexes obtained from cell extracts using TIRF microscopy[8]. Lysates of HeLa cells expressing Halo-tagged Hook3$_{1-552}$ labeled with TMR-HaloTag ligand were flowed into a chamber containing Taxol-stabilized microtubules immobilized on coverslips. The dynein-driven motility of single molecules was then monitored both in the absence or the presence of increasing concentrations of Helix-1 or Helix-1$_{F447A,F448A}$, a peptide carrying the two mutations found to inhibit binding of full-length LIC1 to the Hook domain (Fig. 2d). Consistent with previous reports[5,8], in the absence of Helix-1 we observed robust motility of Halo-Hook3$_{1-552}$-positive complexes along microtubules, characterized by long run lengths and high velocities (Fig. 5a). In contrast, we observed a marked inhibition of processive motility with the addition of Helix-1, with nearly complete inhibition at Helix-1 concentrations of 100 µM or higher, whereas the addition of Helix-1$_{F447A,F448A}$ did not inhibit motility (Fig. 5a). Similar results were observed in experiments that tracked the movement of dynein–dynactin-BICD2 complexes obtained from cell extracts expressing Halo-BICD2$_{1-572}$ labeled with TMR-HaloTag ligand (Fig. 5b). In this case, however, higher concentrations of Helix-1 (>200 µM) were required for full inhibition, which is not entirely unexpected for in trans competition of an intramolecular interaction.

To assess whether the LIC1-effector interaction contributes to organelle motility in cells, we analyzed the distribution of lysosomes in HeLa cells expressing GFP, LIC1$_{WT}$-GFP, or the mutant LIC1$_{F447A,F448A}$-GFP that does not interact with Hook1 (Fig. 2d). Importantly, this mutation is predicted to also block the interaction of LIC1 with other effectors, since we found that Helix-1 is involved in interactions with several effectors (Figs. 2e, g and 4d, e). Lysosomes are well-characterized cargoes of dynein, which drives perinuclear clustering of lysosomes near microtubule minus ends[17,18], and LIC1 is known to be required for this activity[19]. Compared to the expression of GFP alone, the expression of LIC1$_{WT}$-GFP did not significantly change the distribution of lysosomes, visualized by anti-LAMP1 staining of cells fixed 18–22 h after transfection. In contrast, the expression of the LIC1$_{F447A,F448A}$-GFP mutant resulted in an abnormal localization of lysosomes (Fig. 5c). In these cells, lysosomes appeared dispersed throughout the cytoplasm and did not show the characteristic perinuclear clustering seen in control cells (Fig. 5c). In a blind analysis, LIC1$_{F447A,F448A}$-GFP-expressing cells displayed a significantly higher percentage of abnormally positioned lysosomes compared to cells expressing GFP or LIC1$_{WT}$-GFP (Fig. 5d). Together, these results show that the LIC1-effector interaction mediated by Helix-1, and specifically residues F447 and F448, is absolutely required for processive dynein-based motility in vitro and in cells.

## Discussion

Cytoplasmic dynein is responsible for most cellular activities requiring microtubule minus-end-directed motility. However, in isolation, dynein is not processive[5,20]. It is now recognized that dynein's functional diversity, including cargo-specificity and processivity, depends on its interaction with the general adaptor dynactin, regulated by an ever-expanding family of dynein–dynactin effector proteins, including BICD2[5–7], Hook1/3[5,8,9], Spindly[5], FIP3[5], and NIN/NINL[10]. These proteins have been distinctly called adaptors[5,9] or regulators[21]. We have used here the more general term 'effectors' because they do both—they help bring together dynein and dynactin and recruit specific cargoes, which are typical adaptor functions, but they also activate dynein processivity, thus playing a regulatory role. We have a limited understanding of how dynein–dynactin effectors exert these

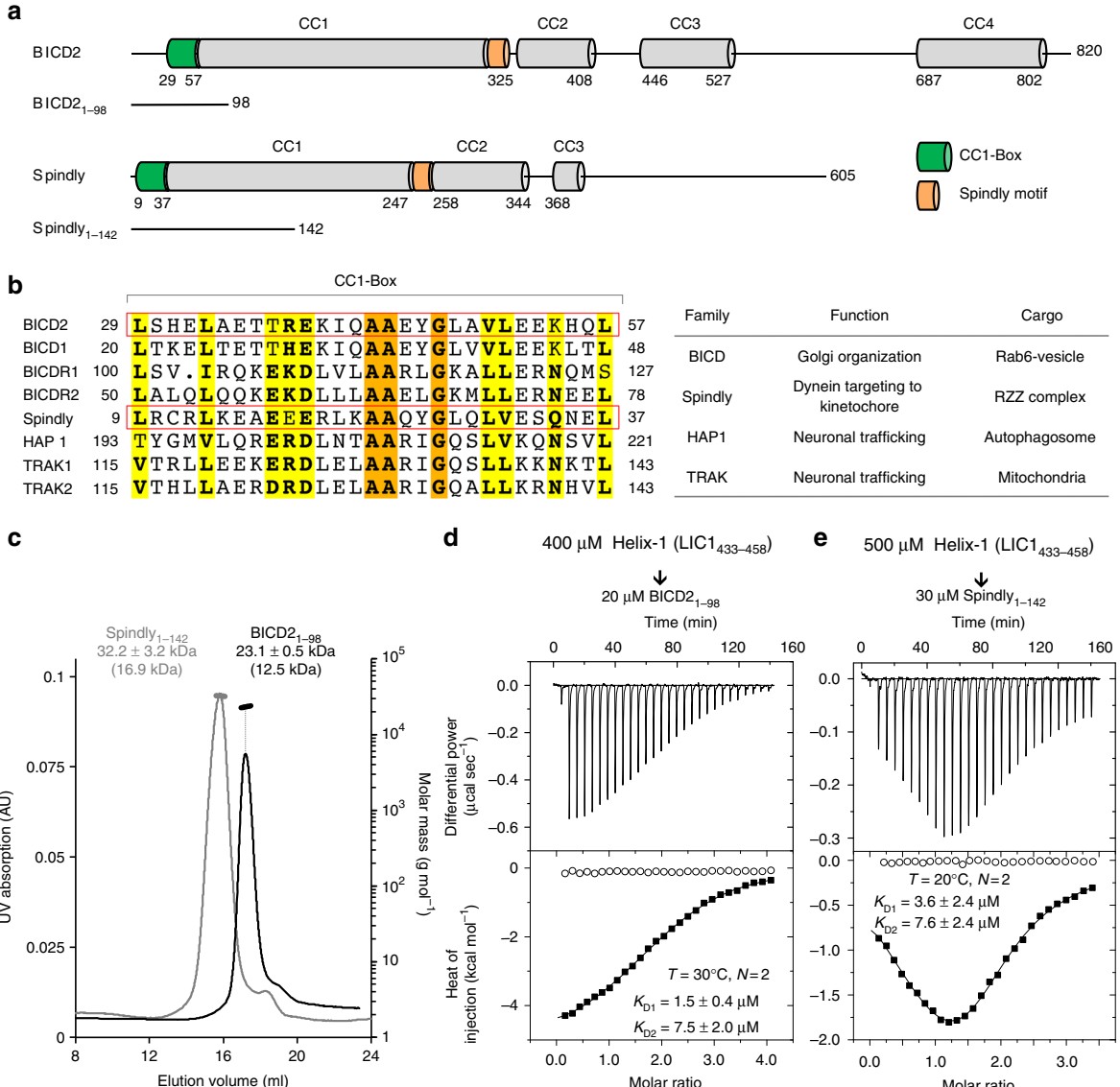

**Fig. 4** LIC1 Helix-1 mediates the interaction with CC1-Box-containing effectors. **a** Domain organization of human BICD2 and Spindly and constructs used in this study. **b** Sequence alignment of the CC1-Box region (left) of several proteins that link dynein to different cargoes and perform different functions[3,42-44] (right). Yellow and orange backgrounds indicate 70% and 100% sequence conservation, respectively. **c** SEC-MALS analysis of BICD2$_{1-98}$ (black) and Spindly$_{1-142}$ (gray). The molar masses determined from light scattering (right y-axis) and the UV absorption at 280 nm (left y-axis) are plotted as a function of the elution volume. The theoretical masses are given in parenthesis. **d**, **e** ITC titrations of Helix-1 (LIC1$_{433-458}$) into BICD2$_{1-98}$ and Spindly$_{1-142}$, respectively. The experimental conditions and fitting parameters are listed for each titration. Errors correspond to the s.d. of the fits. Open symbols correspond to control titrations into buffer

diverse functions, and the lack of recognizable sequence similarity or a common dynein–dynactin-binding motif among all of them has limited our ability to establish general structural-functional correlations. Most of the effectors, however, appear to have cargo-specific binding domains toward their C-termini[12,14,22–25]. Another structural feature shared by all the known effectors is the presence of long regions of coiled-coil. Cryo-EM[16,26–29], biochemical[30–33], and proteomics[10] studies have shown that there are at least three major points of contact among dynein, dynactin, and the effectors, and each of these interactions could in principle contribute toward the overall affinity of their ternary complexes, as well as the adaptor and regulatory activities of each effector. First, dynein and dynactin interact with each other with micromolar affinity ($K_D = \sim3\,\mu M$) via a direct interaction between the dynein intermediate chain and the p150$^{Glued}$ subunit of dynactin[30–33], as well as through the tail domain of the heavy chain[16,26–29]. Second, a long coiled-coil segment in each effector appears to intercalate at the interface between dynein and dynactin, running along the dynactin surface with the N-terminal-end directed toward the barbed end of the actin-like dynactin filament[16,26–29]. This interaction appears to modulate the affinity between dynein and dynactin, as well as the number of dynein molecules that are recruited onto the dynactin scaffold[28,29].

Evidence for the third type of interaction, involving dynein's LIC1 subunit and N-terminal sequences in various effectors, has so far been limited to pull-down[9,11,12] and proteomics[10] studies. Here, we have mapped this interaction to a conserved helix (Helix-1) within the otherwise unstructured and poorly conserved C-terminal region of LIC1. We have further shown that Helix-1 mediates the interaction with structurally and functionally unrelated effectors, and that the interactions typically have low

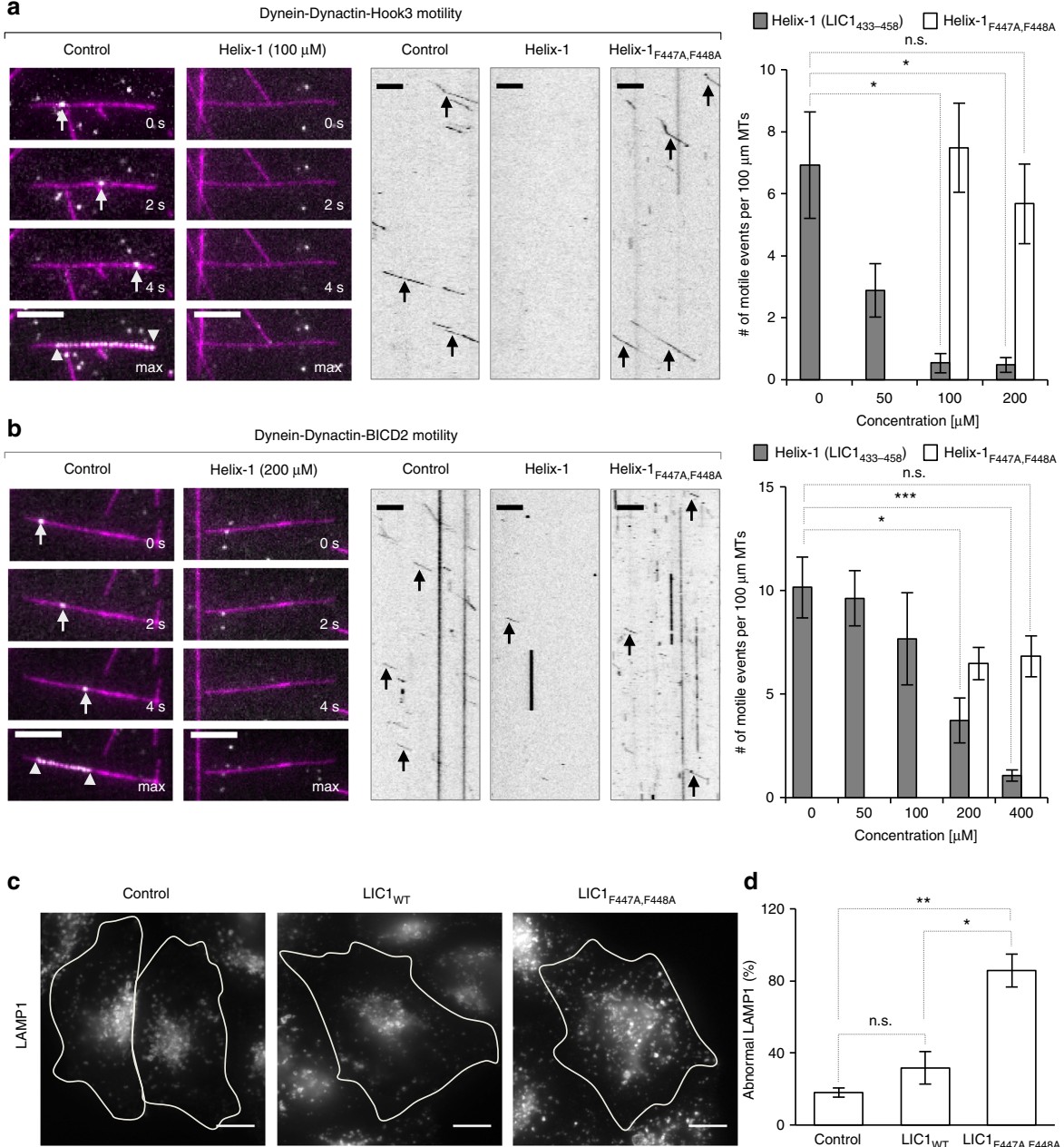

**Fig. 5** The Helix-1-effector interaction is important for processive motility in vitro and in cells. **a**, **b** Time series and kymographs (1 min) of Halo-Hook3$_{1-552}$ and Halo-BICD2$_{1-572}$ runs on microtubules (magenta) in the absence (control) or the presence of Helix-1 or Helix-1$_{F447A,F448A}$ peptides (as indicated) analyzed by TIRF microscopy. Arrows indicate a motile particle and arrowheads indicate the beginning and end of the trajectory in a maximum projection (max). Scale bar, 5 μm. Quantifications (right) show that the number of motile events declines with increasing Helix-1 concentrations, but not Helix-1$_{F447A,F448A}$. The statistical significance of the measurements was determined using a One-way Anova test, analyzing $N = 6$–21 videos and a minimum of 3 individual cell lysates per condition (n.s., non-significant; *$p \leq 0.05$; **$p \leq 0.01$; ***$p < 0.001$). Error bars correspond to the s.e.m. **c** Representative images of LAMP1 staining of fixed HeLa cells expressing GFP, LIC1$_{WT}$-GFP or LIC1$_{F447A,448A}$-GFP. Note that the LAMP1 puncta become more dispersed with the expression of LIC1$_{F447A,448A}$-GFP, but not LIC1$_{WT}$-GFP. Cell perimeters are outlined in white. Scale bar, 10 μm. **d** Percentage of cells with abnormal LAMP1 staining from fixed HeLa cells expressing GFP, LIC1$_{WT}$-GFP and LIC1$_{F447A,448A}$-GFP. The statistical significance of the measurements was determined using a One-way Anova test, analyzing $N = 148$ (GFP), $N = 77$ (LIC1$_{WT}$-GFP), and $N = 48$ (LIC1$_{F447A,448A}$-GFP) cells from three independent repeats (n.s., non-significant; *$p \leq 0.05$; **$p \leq 0.01$). Error bars correspond to the s.e.m

micromolar affinities. We were also able to visualize the structural basis of this interaction at high-resolution for the Hook subfamily of effectors (comprising three isoforms, Hook1–3). Finally, we demonstrated that this interaction enhances the processive motility of dynein in vitro and that disruption of the LIC1-effector interface affects organelle transport in cells. Somewhat analogous to our findings, an interaction between the light and

intermediate chains of yeast dynein has been implicated in dimerization and processive motility[34].

Curiously, the LIC1-effector interaction involves a conserved motif on the LIC1 side of the interface, but different surfaces on the effector side. For Hook-family effectors, the interaction involves the N-terminal Hook domain, which has a globular fold related to the CH domain. However, it is the extended helix α8 of

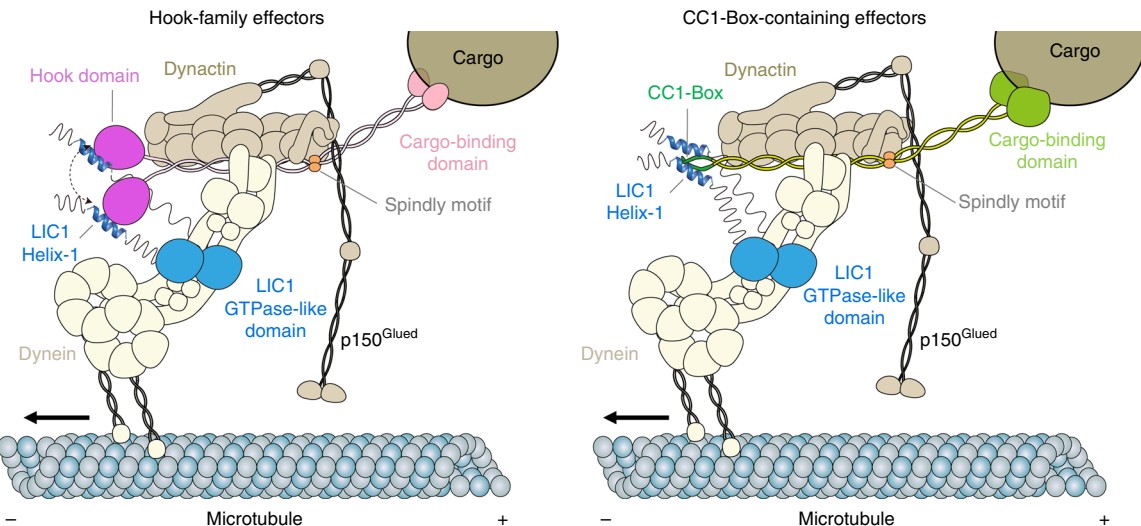

**Fig. 6** Model for cargo transport by dynein–dynactin-effector complexes. Dynein is a 1.4 MDa homodimeric complex of two heavy chains, which each binds smaller subunits, including the intermediate chain, light intermediate chain, and three light chains. Dynactin is a ~1.0 MDa complex of more than 20 proteins, including an actin filament-like core of actin-related protein 1 (Arp1) subunits, capped at both ends by several subunits, and a 'shoulder' domain from which emerges the largest subunit, p150[Glued], which projects ~50 nm and can bind microtubules directly to help initiate a processive run. Although dynein and dynactin bind directly to each other via the dynein intermediate chain and the dynactin p150[Glued] subunit[30–33], they form a stable processive complex only in the presence of effector proteins, including Hook1/3[5,8,9], BICD2[5–7], Spindly[5], FIP3[5], and NIN/NINL[10]. These effectors are unrelated to each other in sequence and recruit different cargoes. Here we have demonstrated that independent of these differences, they all appear to interact with the same region of the dynein LIC1 subunit, which we named Helix-1 (or Effector-Binding Helix, EBH). The interaction involves the Hook domain in Hook-family effectors (left) or a coiled-coil segment in CC1-Box-containing effectors, such as BICD and Spindly (right). We have proposed here that the LIC1-effector interaction may help stabilize the 'parallel-heads' conformation thought to be necessary for dynein processivity[16]

the Hook domain (absent in the CH fold) that mediates the interaction by forming a V-shaped hydrophobic cleft after splitting into two helices. In BICD2 and Spindly, the interaction involves the CC1-Box, which forms part of a longer coiled-coil segment (Fig. 4a). Conceivably, the two helices of the coiled-coil could separate, partially exposing the hydrophobic core of the coiled-coil to create symmetric binding sites for LIC1 Helix-1 on both sides of the coiled-coil. This would give rise to a binding site that is different in sequence, but possibly structurally similar to that of the Hook domain.

The effectors analyzed here bind LIC1 with 1:1 stoichiometry, or rather with 2:2 stoichiometry, since all the effectors identified to date form dimers. In this way, each effector could in principle tether two LIC1 subunits from a single dynein dimer or from two different dynein dimers bound simultaneously to the dynactin complex. In the case of Hook-family effectors, the two LIC1-binding sites are physically separated from one another, as suggested by our rotary shadowing EM analysis (Fig. 1h, i), whereas in CC1-Box-containing effectors the two binding sites occur on the same coiled-coil, i.e., adjacent to each other. Such structural differences, as well as differences in the affinities of the LIC1-effector interactions, may play a modulatory role, by forming dynein–dynactin-effector complexes of different affinities and characterized by different run lengths. In this regard, it is important to note that the dynein–dynactin-Hook3 complex displays a bimodal velocity distribution and faster velocities[8] than the dynactin-BICD2 complex characterized by a single velocity distribution[5,6].

The activation of dynein processivity proceeds through a conformational change from an auto-inhibited so-called 'phi-particle' state to a 'parallel-heads' state capable of binding microtubules upon complex formation with dynactin effectors[16]. The auto-inhibited state is stabilized by inter-heavy chain interactions, including near the LIC1 subunit. Because, the LIC1-effector interaction appears to 'pull' on the LIC1 subunit and with

it on the dynein heavy chain, we propose that it may help reposition the dynein heads for optimal interaction with microtubules by breaking the interaction between heavy chains (Fig. 6), and, thus, this interaction would be specifically important for the activation of dynein processivity. It remains to be demonstrated whether the LIC1-effector interaction is engaged at all times and whether it is absolutely required for dynein–dynactin complex formation, which could primarily depend on the other two interactions mentioned above (i.e., direct dynein–dynactin contacts and contacts mediated by an N-terminal coiled-coil segment of the effectors).

## Methods

**Proteins**. The cDNA encoding for human Hook1 (UniProt: Q9UJC3-1) and Hook3 (UniProt: Q86VS8-1) were purchased from Open Biosystems (Huntsville, AL). Constructs Hook1$_{11–166}$, Hook1$_{11–238}$, and Hook1$_{11–443}$ were cloned between BamHI and SalI sites of vector pMAL-c2x (NEB, Ipswich, MA). All the primers used in cloning are listed in Supplementary Table 1. Construct Hook1$_{1–239}$GCN4 was obtained by adding a 28-a.a. GCN4 sequence (MKQLEDKVEELLSKNYH-LENEVARLKKL) by overlapping primers, while respecting the coiled-coil heptad register. The fusion construct was cloned as above. Constructs Hook3$_{1–143}$ and Hook3$_{1–160}$ were cloned between NotI and SalI sites of a modified pMAL-c2x (NEB) vector in which the Sac1 site after MBP residue N367 was replaced with a NotI site. Point mutations A138D and M140D in Hook3$_{1–160}$ were introduced using the QuikChange site-directed mutagenesis kit (Agilent Technologies, Wilmington, DE). All the proteins were expressed in E. coli BL21 (DE3) cells (Invitrogen, Carlsbad, CA), grown in Terrific Broth medium at 37 °C until the OD$_{600}$ reached a value of 1.5–2, followed by 16 h at 19 °C in the presence of 0.25 mM isopropyl-β-D-thiogalactoside. The cells were collected by centrifugation, resuspended in 20 mM Tris, pH 7.0, 100 mM NaCl, 4 mM benzamidine hydrochloride, 1 mM PMSF, and 1 mM DTT and lysed using a Microfluidizer large-scale homogenizer (Microfluidics, Newton, MA). All the proteins were purified through an amylose affinity column according to the manufacturer's protocol (NEB). The MBP tag was removed by incubation with TEV protease overnight at 4 °C. The proteins were additionally purified by gel filtration on a SD200HL 26/60 column (GE Healthcare, Little Chalfont, UK) in 20 mM Tris, pH 7.0, 100 mM NaCl, 1 mM DTT.

The cDNA encoding for human LIC1 (UniProt: Q9Y6G9-1) was a generous gift from Ronald Vale (UCSF). Constructs LIC1$_{FL}$, LIC1$_{1–461}$, and LIC1$_{1–437}$ were

amplified by PCR and cloned between the BamHI and SalI sites of a modified-vector pMAL-c2x that adds a C-terminal Strep-tag to the target protein. Point mutations F447A and F448A in LIC1$_{FL}$ were introduced using the QuikChange site-directed mutagenesis kit. Proteins were expressed and purified as described above, with one exception; after amylose affinity purification, the proteins were loaded onto a StrepTactin Sepharose column (IBA Lifesciences, Göttingen, Germany) and eluted after extensive washing with 3 mM desthiobiotin, 20 mM Tris, pH 7.0, 100 mM NaCl, and 1 mM DTT. To obtain the Helix-1 peptides, the cDNA encoding for LIC1$_{433–458}$ (Fig. 2a) was cloned between the SapI and SalI sites of vector pTYB11 (NEB). Point mutations F447A and F448A were introduced using the QuikChange site-directed mutagenesis kit to obtain the mutant peptide Helix-1$_{F447A,F448A}$. Proteins were expressed as above, and purified on a chitin affinity column according to the manufacturer's protocol (NEB), followed by auto-cleavage of the intein tag induced by incubation with 50 mM DTT overnight at 4 °C. The cleaved peptides were additionally purified on a Symmetry300 C$_{18}$ reverse-phase column (Waters, Milford, MA) using an acetonitrile gradient of 0–90% (v/v) and 0.1% (v/v) trifluoroacetic acid.

The cDNA encoding for full-length human Hook1 was codon optimized for expression in Sf9 cells and synthesized (Genscript Biotech, Piscataway, NJ). The gene was cloned between the SalI and XbaI sites of a modified-vector pFastBac1, which adds a V5 epitope tag at the N terminus and a Strep-tag at the C-terminal of the target protein. The protein was expressed in *Spodoptera frugiperda* 9 (Sf9) cells using the Bac-to-Bac baculovirus expression system according to the manufacturer's protocol (Invitrogen). The cells were collected by centrifugation, re-suspended in lysis buffer (10 mM Na$_2$HPO$_4$, pH 7.4, 100 mM NaCl, 1 mM PMSF, 4 mM Benzamidine, 1 mM DTT, and 5% glycerol (v/v)) with addition of a protease inhibitor cocktail (Roche, Basel, Switzerland). The cells were lysed by addition of 0.5% (v/v) Triton X-100 through three cycles of freeze–thaw on ice and centrifuged 20 min at 20,000 × g. Lysates were loaded onto a StrepTactin Sepharose column, and after washing extensively with lysis buffer, Hook1$_{FL}$ was eluted with the addition of 3 mM desthiobiotin. The protein was additionally purified through a Superose 6 gel filtration column (GE Healthcare), equilibrated with 20 mM Tris, pH 7.5, 100 mM NaCl, and 1 mM DTT.

The cDNAs encoding for human BICD2$_{1–98}$ (UniProt: Q8TD16-1) and Spindly$_{1–142}$ (UniProt: Q96EA4-1) were synthesized with codon optimization for *E. coli* expression (Genscript Biotech). The BICD2$_{1–98}$ gene was cloned between the BamHI and SalI sites of vector pMAL-c2x. The protein was expressed and purified as described above. The Spindly$_{1–142}$ gene was cloned between the BamHI and SalI sites of vector pCold1 (TAKARA BIO, Kusatsu, Japan). The protein was expressed in BL21 (DE3) cells as described above. The protein was purified through a Ni-NTA affinity column in 50 mM Tris, pH 8.0, 500 mM NaCl, 4 mM benzamidine hydrochloride, and 1 mM PMSF and eluted with 250 mM Imidazole. The His$_6$-tag was removed by overnight incubation with TEV protease at 4 °C, and the protein was additionally purified through a HiLoad 16/600 Superdex 75 pg column (GE Healthcare) in 20 mM Tris pH 7.0, 100 mM NaCl, and 1 mM DTT.

**Multi-angle light scattering**. Samples were separated by size exclusion chromatography on a Superose 6 10/300 GL column (GE Healthcare) equilibrated with 20 mM Tris, pH 7.5, 100 mM NaCl, and 1 mM DTT, using an Agilent 1100 HPLC system (Agilent Technologies). Multi-angle light scattering was measured in line using a DAWN-HELEOS Multi-angle light scattering detector and an Optilab rEX refractive index detector. The scattering data were analyzed with the ASTRA software (Wyatt Technology, Santa Barbara, California).

**Isothermal titration calorimetry**. ITC measurements were carried out on a VP-ITC instrument (MicroCal, Northampton, MA). Protein samples were dialyzed for 2 d against 20 mM HEPES, pH 7.5, 100 mM NaCl, and 0.25 mM TCEP (ITC buffer). The LIC1$_{433–458}$ peptide was re-suspended in ITC buffer, followed by three cycles of lyophilization/resolubilization in 50% (v/v) methanol to remove any tri-fluoroacetic acid remaining after reverse-phase purification. The peptide was then re-suspended in ITC buffer. The proteins (or LIC1$_{433–458}$ peptide) in the syringe were titrated at a concentration 10− to 20−fold higher than that of the proteins in the ITC cell of total volume 1.44 ml (as indicated in the figures). Titrations consisted of 10 µl injections, lasting for 10 s, with an interval of 300–400 s between injections. The heat of binding was corrected for the heat of injection, determined by injecting proteins into buffer. Data were analyzed using the program Origin (OriginLab, Northampton, MA). The temperature and parameters of the fit (stoichiometry and affinity) of each experiment are given in the figures.

**Rotary shadowing and electron microscopy**. Full-length Hook1 was suspended in a solution containing 10 mM Tris, pH 7.5, 50 mM NaCl, 1 mM DTT, and 50% (v/v) glycerol. Samples were diluted to a concentration of 100 µg/ml in the same solution and 1 µg of each sample were sprayed on a freshly split mica surface and dried for 1 h at room temperature. The samples were rotary shadowed with platinum at a 7° angle, and replicated with carbon in a Balzers 410 freeze-fracture machine. Replicas were photographed at a magnification of 98,900 using a Philips 410 transmission electron microscope operating at 80 kV. The original images were obtained from areas at the edge of each droplet that showed distinct non-aggregated molecules and a clear background. Images were analyzed using the ImageJ software[35].

**Crystallization and structure determination**. Hook3$_{1–160}$ at 10 mg/ml in 10 mM Tris, pH 7.4, 25 mM NaCl, and 2 mM TCEP was mixed with 1.2 molar excess of the LIC1$_{433–458}$ peptide at 4 °C for 1 h. Crystal were obtained at 20 °C using the hanging-drop method. The crystallization drop consisted of a 1:1 (v/v) mixture of protein solution and well solution (1.44 M ammonium citrate tribasic, pH 6.25). The crystals were improved through consecutive rounds of micro-seeding. For data collection, the crystals were flash-frozen in liquid nitrogen from a cryo-solution consisting of crystallization buffer with addition of 30% (v/v) glycerol.

An x-ray diffraction dataset was collected at the Cornell High Energy Synchrotron Source (CHESS) beamline F1. The diffraction data were indexed and scaled using the program HKL2000[36]. A molecular replacement solution was obtained with the program Phenix[37] using PDB entry 5J8E (unbound Hook domain of Hook3[9]). Model building and refinement were carried out with the programs Coot[38] and Phenix[37]. Figures were generated with the program PyMOL (Schrödinger, New York City, NY). Sequence alignments were carried out with the program MAFFT[39] and visualized using ESPript[40]. Data collection and refinement statistics are listed in Table 1.

**Single-molecule motility assays**. The motility of dynein–dynactin-Hook3$_{1–552}$ or dynein–dynactin-BICD2$_{1–572}$ complexes from cell extracts were tracked using TIRF microscopy[8]. The motility assays were performed in flow chambers constructed with a glass slide and a silanized (PlusOne Repel Silane, GE Healthcare) coverslip, held together with double sided adhesive tape and vacuum grease to form a ~15 µl chamber. A 1:40 dilution of monoclonal anti-β-tubulin antibody (T5201, Sigma) was perfused into the chamber, which was subsequently blocked with 5% pluronic F-127 (Sigma-Aldrich, St. Louis, MO). Taxol-stabilized microtubules, labeled with HiLyte 488 or 647 (Cytoskeleton, Denver, CO) at a labeling ratio of 1:40, were flowed into the chamber and immobilized by interaction with anti-β-tubulin antibodies.

HeLa cells expressing Halo-tagged Hook3$_{1–552}$ or BICD2$_{1–572}$ were labeled with TMR-HaloTag ligand (Promega, Madison, WI) 18–20 h post-transfection. Cells grown in 10 cm plates to 70–80% confluence were then lysed in 100 µl lysis buffer (40 mM HEPES pH 7.4, 120 mM NaCl, 1 mM EDTA, 1 mM Mg-ATP, 0.1% Triton X-100, 1 mM PMSF, 0.01 mg/ml TAME, 0.01 mg/ml leupeptin and 1 µg/ml pepistatin-A). Cell lysates were then clarified by centrifugation at 17,000× g. Before flowing into the imaging chamber, the cell extracts were diluted in P12 buffer (12 mM PIPES, pH 6.8, 1 mM EGTA, 2 mM MgCl$_2$, and 20 µM Taxol). The cells lysates were then further diluted in motility buffer (1 × P12 buffer supplemented with 10 mM Mg-ATP, 0.3 mg/ml casein, 0.3 mg/ml bovine serum albumin, and 10 mM DTT) and an oxygen scavenging system (0.5 mg/ml glucose oxidase, 470 U/ml catalase and 15 mg/ml glucose) and flowed into the chamber to be imaged.

The dynein-driven motility of single Halo-Hook3$_{1–552}$ or Halo-BICD2$_{1–572}$ positive molecules was then examined in the absence or the presence of varying concentrations of the Helix-1 (or Helix-1$_{F447A,F448A}$) peptide, added to the motility buffer immediately prior to the addition of the cell lysates. All the movies (4 frames/s) were acquired at room temperature using a Nikon TIRF system (Perkin Elmer, Waltham, MA) on an inverted Ti microscope equipped with a 100 × objective and an ImageEM C9100-13 camera (Hamamatsu Photonics, Hamamatsu City, Japan) controlled by Volocity software (Improvision, Lexington, MA). Particle tracking was performed using the TrackMate plugin in the program Fiji[41]. Particle runs were tracked only if both the start and end of a run were observable over the course of the movie. Runs on microtubule bundles were excluded from this analysis.

**Lysosomal distribution analysis**. HeLa cells transfected for 18–22 h with GFP, LIC1$_{WT}$-GFP, or LIC1$_{F447A,F448A}$-GFP were fixed with PFA. Cells were then stained with anti-LAMP1 antibodies (rabbit polyclonal, ab24170, Abcam, Cambridge, UK) diluted 1:500, followed by anti-rabbit secondary antibodies, and mounted on glass coverslips with ProLong Gold anti-fade reagent (Invitrogen). Images were taken with an inverted epifluorescence microscope (DMI6000B, Leica Camera AG, Wetzlar, Germany) using an Apochromat 63 ×, 1.4 NA oil immersion objective (Leica Camera AG).

The images were blinded before analysis, and the distribution of LAMP1-positive vesicles was scored using cells expressing GFP only. LAMP1 staining was scored "abnormal" if the cells lacked the characteristic perinuclear lysosomal clustering and/or had several or a few enlarged LAMP1-positive vesicles dispersed throughout the cell.

**Data availability**. Atomic coordinates and structure factor amplitudes for the crystal structure of the Hook3-LIC1 complex were deposited with the Protein Data Bank (PDB) under accession code 6B9H. Other data and materials are available from the corresponding author upon reasonable request.

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

## Acknowledgements

This work was supported by National Institutes of Health Grants P01 GM087253 to R.D. (Project 3) and E.L.F.H. (Project 4) and R01 GM073791 to R.D.

## Author contributions

I.-G.L., E.L.F.H., and R.D. designed the experiments. I.-G.L. performed the biochemical and structural experiments with M.B. and R.D.'s participation. M.A.O. performed the in vitro motility and cellular experiments. C.F.-A. performed the rotary shadowing EM experiments. I.-G.L. and R.D. wrote the manuscript. All the authors reviewed the figures and manuscript and approved its final version.

## Additional information

**Competing interests:** The authors declare no competing interests.

