## [Peer Review File · Nature Communications]

Reviewers' comments:

Reviewer #1 (Remarks to the Author):

This manuscript provides significant new insight into how cargo adaptors recruit dynein and dynactin. The authors answer major questions about the location and nature of the interaction between the dynein light intermediate chain 1 (LIC1) and various cargo effectors. These cargo effectors recruit dynein/dynactin to cargo and activate the molecular machine. The strength of this manuscript is the structural data presented and also the careful work they do to localise and quantify the interaction between LIC1 and the effectors, primarily using isothermal calorimetry (ITC). The results will be of great interest to those in the molecular motor field.

The authors use ITC to localise the Hook:LIC1 interaction and quantify the stoichiometry and strength of the interaction. They then visualise Hook1 using rotary shadowing EM, and assign aspects of its structure. The manuscript next describes experiments to localise the interaction within the LIC1 C-terminus, presenting good data to show helix-1 from LIC1 is responsible for the interaction. An exciting structure is next presented of a Hook domain in complex with the interacting helix from LIC1, showing the atomic details of the interaction. This structure shows that in contrast to a previous structure of the Hook domain alone, one helix is rearranged to accommodate the LIC1 interaction. This structure is followed by careful experiments to validate the interaction site. Next, the authors show that the same region of the LIC1 interacts with both BICD2 and Spindly using ITC. These effectors share sequence similarity with other adaptors via a previously-defined CC1-Box. These data strongly support the idea that this region of the LIC is important for interactions with many varied dynein/dynactin effectors. Finally, the authors present important evidence that this LIC region is necessary for dynein/dynactin motility of different cargoes in vitro and in vivo. They do this by inhibiting motility in both cases using a LIC1433-458 peptide. Overall, the data presented here represent strong evidence to show the nature and importance of the LIC1/effector interaction for dynein/dynactin complex motility. The work is well executed and I strongly recommend its publication. I have a few suggestions for improvements:

Major points:

- 1) Some of the ITC experiments are performed at 30°C (Fig 1d, Fig 4d), whereas others are performed 20°C. Does the difference in temperature have an effect on the KD?
- 2) The ITC curve in Figure 1d does not appear to saturate at high molar ratio. Are there additional data points missing? Could the authors comment on this?
- 3) The ITC curve in Figure 4d (Spindly, right panel) decreases at low molar ratio, then increases at around 1.25 molar ratio. Can the authors address this curve shape?
- 4) The concentration of peptide required to inhibit dynein motility in cell lysates is high (100uM) which raises the question of whether the effect is specific? Could the authors use a peptide with mutated phenylalanines as a control. One would predict that it has no inhibition at the same concentration.

Minor points:

- 1) MBP-LIC1-454 binds with lower affinity than MBP-LIC-FL. If the authors have data with MBP-LIC1-458 (which should have the same affinity as MBP-LIC-FL) then it would be great if they could include it. I don't think it is an essential point as there is other data to suggest helix-1 is the site of interaction. It would be useful if the authors could state in this section that their data suggest helix-2

is not required and somewhere discuss what the role of this conserved helix might be.

2) Please state the number of events picked in Fig 5c and the number of cells analysed in Figure 5e in the legend.

3) Please include the associated error in the MW measurements in the SEC-MALS data in Figure 1c and Figure 4c.

4) Please report N, the number of binding sites, if possible in figure 4d.

5) The source or synthesis method of the LIC1433-458 peptide was absent in the methods. Could the authors include this?

6) In Figure 1 and Figure 4 the predicted CC of Hook, BICD2 and Spindly are shown. Low resolution structures by cryo-EM of BICD2 and Hook suggest the whole region between CC1 and the spindly motif are probably continuous coiled coil.

7) Could the authors include kymographs in Figure 5a/b?

Reviewer #2 (Remarks to the Author):

The manuscript by Lee et al. provides new insights into the molecular function of the light intermediate chain (LIC) of the microtubule motor complex, cytoplasmic dynein-dynein-Hook. The authors reveal that a conserved helix within the C-terminus of LIC interacts with the N-terminus of Hook, and that the N-terminus of Hook undergoes a conformational change upon its interaction with the LIC, permitting the binding of LIC. Specifically, Lee et al. show that the interaction between LIC and Hook is mediated by highly conserved hydrophobic residues in Helix 1 of the LIC that are inserted into a hydrophobic cleft formed at the N-terminus of Hook. In addition, the authors demonstrate that Helix 1 of LIC is a common binding site for other dynein effectors, such as the CC1-Box-containing cargo-adaptors/ effectors BicD and Spindly. Finally, Lee et al. show that the newly discovered interaction between LIC and Hook is not only required for the ability of dynein-dynein-Hook complex to move processively along microtubules *in vitro*, but also for its functions in cells. The reported results are highly relevant for the cell biology and cytoskeletal communities. The manuscript is written with care and I feel that it will be a timely and well-cited contribution. I thus highly recommend its publication by Nature Communications.

As the study is carried out with great care and rigor, I only have a minor comment:

The finding that the interaction between the LIC and Hook is required for the processive motion of dynein is intriguing. While the authors provide a possible explanation for how this could be achieved by writing that the LIC-Hook interaction could relieve dynein's auto-inhibited conformation by "repositioning the dynein heads for optimal interactions with microtubules", I recommend that the authors also discuss the possibility that the interaction of the LIC with the dynein heavy chain (HC) could have similar effects on dynein processivity as the interaction of dynein's intermediate chain (IC) with the dynein HC: A recent study has shown that the light chain (LC)-IC complex promotes dynein HC dimerization and potentiates processivity (Rao et al., *Mol. Biol. Cell*, 2013). It could therefore be that the LIC-Hook interaction not only contributes to the activation of dynein but also changes how the LICs interact with the dynein HCs, and that this change in LIC-HC interaction results in the HC dimerization at a location more proximal to the motor domains. The latter effect could improve head-head coordination and therefore increase dynein processivity. I recommend discussing this possibility.

Response to Reviewers (Blue text)

The reviewers were very enthusiastic. Nevertheless, we followed every single one of their recommendations, as we found that they could improve the paper. We are very thankful for their comments.

Note further that in the interim, we improved the resolution of the structure of the complex between the Hook domain and the dynein LIC1 helix-1 from 2.4 Å (collected at our home source) to 1.5 Å (collected at a synchrotron beamline). The revised manuscript now describes the higher-resolution structure.

Reviewer 1

This manuscript provides significant new insight into how cargo adaptors recruit dynein and dynactin. The authors answer major questions about the location and nature of the interaction between the dynein light intermediate chain 1 (LIC1) and various cargo effectors. These cargo effectors recruit dynein/dynactin to cargo and activate the molecular machine. The strength of this manuscript is the structural data presented and also the careful work they do to localise and quantify the interaction between LIC1 and the effectors, primarily using isothermal calorimetry (ITC). The results will be of great interest to those in the molecular motor field.

The authors use ITC to localise the Hook:LIC1 interaction and quantify the stoichiometry and strength of the interaction. They then visualise Hook1 using rotary shadowing EM, and assign aspects of its structure. The manuscript next describes experiments to localise the interaction within the LIC1 C-terminus, presenting good data to show helix-1 from LIC1 is responsible for the interaction. An exciting structure is next presented of a Hook domain in complex with the interacting helix from LIC1, showing the atomic details of the interaction. This structure shows that in contrast to a previous structure of the Hook domain alone, one helix is rearranged to accommodate the LIC1 interaction. This structure is followed by careful experiments to validate the interaction site. Next, the authors show that the same region of the LIC1 interacts with both BICD2 and Spindly using ITC. These effectors share sequence similarity with other adaptors via a previously-defined CC1-Box. These data strongly support the idea that this region of the LIC is important for interactions with many varied dynein/dynactin effectors. Finally, the authors present important evidence that this LIC region is necessary for dynein/dynactin motility of different cargoes in vitro and in vivo. They do this by inhibiting motility in both cases using a LIC1433-458 peptide. Overall, the data presented here represent strong evidence to show the nature and importance of the LIC1/effector interaction for dynein/dynactin complex motility. The work is well executed and I strongly recommend its publication. I have a few suggestions for improvements:

Major points:

1. Some of the ITC experiments are performed at 30°C (Fig 1d, Fig 4d), whereas others are performed 20°C. Does the difference in temperature have an effect on the KD?

Generally, thermodynamic parameters are indeed linked ($\Delta G = -RT \ln K$). Two ITC experiments in the paper (Figs 1d and 4d) were performed at 30°C, because at 20°C (the temperature used for most experiments in the paper) the amount of heat given off

by these two reactions was too small. Therefore, these transitions were hard to resolve above the background, increasing the error in the resulting thermodynamic parameters from fitting (Salim & Feig *Methods* 2009). Moreover, at 20°C the titration in Fig. 4d switches from exothermic to endothermic, making fitting unreliable, particularly for the second binding site (i.e. fitting errors are thus too high). We show here side-by-side the 20°C and 30°C titrations to clarify this point for the reviewer, and also added a clarification in the main text (page 2).

Related to Fig. 1d. Side-by-side comparison of the 20°C and 30°C titrations. Note that the K_D s are similar despite the greater error observed with the 20°C titration.

Related to Fig. 4d. Side-by-side comparison of the 20°C and 30°C titrations. Note that the K_D s for site 1 are similar, but not for site 2 (endothermic part of the reaction).

2. The ITC curve in Figure 1d does not appear to saturate at high molar ratio. Are there additional data points missing? Could the authors comment on this?

We repeated this experiment to address this concern. To explain, even at 30°C (see point 1 above) this titration had low signal/noise ratio. To improve the signal/noise ratio, we increased the concentration of the reactants as much as possible. The highest concentration attainable with MBP-LIC (102 kDa) in the syringe was 400 μ M (~40 mg/ml), compared to 240 μ M in the original submission. This improved saturation (see Fig. 1d above and in the main text). Note, however, that the binding affinity changed minimally.

3. The ITC curve in Figure 4d (Spindly, right panel) decreases at low molar ratio, then increases at around 1.25 molar ratio. Can the authors address this curve shape?

In this titration, the LIC1 peptide binds to two sites in the Spindly coiled coil. The important thing to note is that because the two binding sites have similar affinities, they become saturated at the same time. The first site has a mild endothermic character, whereas the second site has a strong exothermic character, which masks the endothermic signal for the first part of the titration. Additionally, this reaction could be accompanied by a conformational change, which we do not know, but ongoing structural studies in our lab should clarify this point. Clarifications were added in the text to explain this reaction (page 5).

4. The concentration of peptide required to inhibit dynein motility in cell lysates is high (100 μ M) which raises the question of whether the effect is specific? Could the authors use a peptide with mutated phenylalanines as a control. One would predict that it has no inhibition at the same concentration.

We expressed the mutant peptide and added this control. As expected the mutant peptide did not inhibit motility (expanded Fig. 5 and text). Note, however, that the high concentration needed for inhibition is not unexpected, because we are titrating this peptide in *trans* to inhibit an interaction between dynein and dynactin occurring intramolecularly. A note was added in the text to explain this reaction (page 6). Experimental details were also added in the Methods section.

Minor points:

1. MBP-LIC1-454 binds with lower affinity than MBP-LIC-FL. If the authors have data with MBP-LIC1-458 (which should have the same affinity as MBP-LIC-FL) then it would be great if they could include it. I don't think it is an essential point as there is other data to suggest helix-1 is the site of interaction. It would be useful if the authors could state in this section that their data suggest helix-2 is not required and somewhere discuss what the role of this conserved helix might be.

To address this question, we expressed a new construct (MBP-LIC1-461), slightly extending the LIC1 construct to more generously include helix-1. We repeated the binding experiment with MBP-LIC1-461 and, as expected, the affinity was found to be the same as for full-length LIC1, demonstrating that the interaction is fully contained within Helix-1 (revised Fig. 2b). This also demonstrates that helix-2 is not involved in this interaction, although its conservation suggests it may have a different role.

2. Please state the number of events picked in Figure 5c and the number of cells analyzed in Figure 5e in the legend.

This information was added as requested (Legend to Fig. 5).

3. Please include the associated error in the MW measurements in the SEC-MALS data in Figure 1c and Figure 4c.

This information was added as requested (Figs. 1c and 4c).

4. Please report N, the number of binding sites, if possible in figure 4d.

This information was added as requested (Fig. 4d).

5. The source or synthesis method of the LIC1433-458 peptide was absent in the methods. Could the authors include this?

This information was added as requested (Methods).

6. In Figure 1 and Figure 4 the predicted CC of Hook, BICD2 and Spindly are shown. Low resolution structures by cryo-EM of BICD2 and Hook suggest the whole region between CC1 and the spindly motif are probably continuous coiled coil.

Indeed, these coiled coils appear to be stabilized in the complex with dynein and dynactin, since they are clearly interrupted at the sequence level. We changed the diagrams for BICD2 and Spindly (Fig. 4a), but not for Hook (Fig. 1a), since our rotary shadowing, ITC and MALS studies show a break in between the CC1 and CC2 regions of Hook (Fig. 1).

7. Could the authors include kymographs in Figure 5a/b?

This information was added as requested (expanded Fig. 5).

Reviewer 2

The manuscript by Lee et al. provides new insights into the molecular function of the light intermediate chain (LIC) of the microtubule motor complex, cytoplasmic dynein-dynein-Hook. The authors reveal that a conserved helix within the C-terminus of LIC interacts with the N-terminus of Hook, and that the N-terminus of Hook undergoes a conformational change upon its interaction with the LIC, permitting the binding of LIC. Specifically, Lee et al. show that the interaction between LIC and Hook is mediated by highly conserved hydrophobic residues in Helix-1 of the LIC that are inserted into a hydrophobic cleft formed at the N-terminus of Hook. In addition, the authors demonstrate that Helix-1 of LIC is a common binding site for other dynein effectors, such as the CC1-Box-containing cargo-adaptors/ effectors BicD and Spindly. Finally, Lee et al. show that the newly discovered interaction between LIC and Hook is not only required for the ability of dynein-dynein-Hook complex to move processively along microtubules in vitro, but also for its functions in cells. The reported results are highly relevant for the cell biology and cytoskeletal communities. The manuscript is written with care and I feel that

it will be a timely and well-cited contribution. I thus highly recommend its publication by Nature Communications.

As the study is carried out with great care and rigor, I only have a minor comment: The finding that the interaction between the LIC and Hook is required for the processive motion of dynein is intriguing. While the authors provide a possible explanation for how this could be achieved by writing that the LIC-Hook interaction could relieve dynein's auto-inhibited conformation by "repositioning the dynein heads for optimal interactions with microtubules", I recommend that the authors also discuss the possibility that the interaction of the LIC with the dynein heavy chain (HC) could have similar effects on dynein processivity as the interaction of dynein's intermediate chain (IC) with the dynein HC: A recent study has shown that the light chain (LC)-IC complex promotes dynein HC dimerization and potentiates processivity (Rao et al., Mol. Biol. Cell, 2013). It could therefore be that the LIC-Hook interaction not only contributes to the activation of dynein but also changes how the LICs interact with the dynein HCs, and that this change in LIC-HC interaction results in the HC dimerization at a location more proximal to the motor domains. The latter effect could improve head-head coordination and therefore increase dynein processivity. I recommend discussing this possibility.

As requested, we have added this interesting parallel to the Discussion (page 7).

REVIEWERS' COMMENTS:

Reviewer #1 (Remarks to the Author):

The authors have addressed all of my comments. The paper is very carefully put together and makes a really important contribution to the field. I recommend its immediate publication.